# Genetic Subtypes and Natural Resistance Mutations in HCV Genotype 4 Infected Saudi Arabian Patients

**DOI:** 10.3390/v13091832

**Published:** 2021-09-14

**Authors:** Mariantonietta Di Stefano, Mona H. Ismail, Thomas Leitner, Giuseppina Faleo, Saada A. Elmnan Adem, Mohamed O. M. E. Elamin, Obeidi Eltreifi, Marwan J. Alwazzeh, Jose R. Fiore, Teresa A. Santantonio

**Affiliations:** 1Department of Clinical and Experimental Medicine, Section of Infectious Diseases, University of Foggia, 71122 Foggia, Italy; giuseppina.faleo@unifg.it (G.F.); jose.fiore@unifg.it (J.R.F.); teresa.santantonio@unifg.it (T.A.S.); 2College of Medicine, Imam Abdulrahman Bin Faisal University, Dammam 32210, Saudi Arabia; moismail@iau.edu.sa (M.H.I.); moElamin@iau.edu.sa (M.O.M.E.E.); oeltreifi@iau.edu.sa (O.E.); mjalwazzeh@iau.edu.sa (M.J.A.); 3Division of Gastroenterology, King Fahd Hospital of the University, Al-Khobar 34217, Saudi Arabia; sadem@iau.edu.sa; 4Theoretical Biology & Biophysics Group, Los Alamos National Laboratory, Los Alamos, NM 87544, USA; tkl@lanl.gov; 5Department of Biochemistry, King Fahd Hospital of the University, Al-Khobar 34217, Saudi Arabia; 6Department of Microbiology and Laboratory Medicine, King Fahd Hospital of the University, Al-Khobar 34217, Saudi Arabia; 7Infectious Disease Division, King Fahd Hospital of the University, Al-Khobar 34217, Saudi Arabia

**Keywords:** HCV, genotypes, subtypes, DAA

## Abstract

This study aimed to characterize the HCV genetic subtypes variability and the presence of natural occurring resistance-associated substitutions (RASs) in Saudi Arabia patients. A total of 17 GT patients were analyzed. Sequence analysis of NS3, NS5A, and NS5B regions was performed by direct sequencing, and phylogenetic analyses were used to determine genetic subtypes, RAS, and polymorphisms. Nine patients were infected by GT 4a, two with GT 4o and three with GT 4d. Two patients were infected with apparent recombinant virus (4a/4o/4a in NS3/NS5A/NS5B), and one patient was infected with a previously unknown, unclassifiable, virus of GT 4. Natural RASs were found in six patients (35%), including three infected by GT 4a, two by GT 4a/GT 4o/GT 4a, and one patient infected by an unknown, unclassifiable, virus of GT 4. In particular, NS3-RAS V170I was demonstrated in three patients, while NS5A-RASs (L28M, L30R, L28M + M31L) were detected in the remaining three patients. All patients were treated with sofosbuvir plus daclatasvir; three patients were lost to follow-up, whereas 14 patients completed the treatment. A sustained virological response (SVR) was obtained in all but one patient carrying NS3-RAS V170I who later relapsed. GT 4a is the most common subtype in this small cohort of Saudi Arabia patients infected with hepatitis C infection. Natural RASs were observed in about one-third of patients, but only one of them showed a treatment failure.

## 1. Introduction

Hepatitis C virus (HCV) is a leading cause of cirrhosis, hepatocellular carcinoma, liver transplantation, and liver-related death worldwide [1]. The epidemic caused by HCV affects all regions, with significant differences between and within countries. Globally, an estimated 71 million people are chronically infected by HCV. The WHO Eastern Mediterranean Region and the European Region have the highest reported prevalence of HCV [1,2,3].

HCV exhibits high genetic diversity and is currently classified into eight genotypes (GT 1 to GT 8), with varied geographic prevalence [4].

GT 1 is prevalent in North and South America and Western and Northern Europe [5]; GT 2 circulates in Japan and part of Europe and America; GT 3 is widely spread in South Asia, Australia, and Europe [5]. However, the distribution of HCV genotypes in the Middle East is highly variable, with a higher prevalence of GT 1 in Iran, Oman, and UAE. In contrast, GT 2 is prevalent in Bahrain and Libya, GT 3 is prevalent in Afghanistan, and GT 4 is prevalent in Egypt, Iraq, Jordan, Palestine, Qatar, Saudi Arabia, and Syria.

In Saudi Arabia, GT 4 is the prevalent genotype [6]; however, GT 4 subtypes are still poorly investigated. In addition, even fewer data are available on the presence of naturally occurring RAS and their possible impact on the response to treatment with direct-acting antivirals (DAAs). Recently, the natural presence of RASs in the viral genome (NS3, NS5A, and NS5B) was associated with a decreased virological response rates to DAA [7,8,9,10,11,12,13].

This study aimed to characterize the genetic subtypes of HCV-GT4 in Saudi Arabian patients and analyze the prevalence and characteristics of resistance-associated substitutions (RASs) in DAA-naïve patients by the use of Sanger sequencing with a RAS detection limit of around 15/20% of the viral population. The impact of natural RASs on the response to DAA treatment was also evaluated.

## 2. Material and Methods

### 2.1. Study Population

A total of 17 HCV-infected attending the Hepatology Clinic patients at Gastroenterology Section at the King Fahad Hospital of the University, Al Khobar, Saudi Arabia, were included in the study. Of the 17 patients, twelve patients were Saudi, three were Egyptian, one was Sudanese, and one was Palestinian. All patients infected with HCV with detectable HCV RNA were included, and we excluded coinfection with HBV and HIV.

Eleven patients were males and six were females; the mean age was 49 years (range 32 to 73 years). The risk factors included blood transfusion in 8/17 patients (47%), surgery in 1/17 patients (6%), intrafamiliar transmission in 1/17 patients (6%), while in the remaining seven patients, the risk factor was unknown (41%). All patients were infected with GT4 and were DAA-naïve. At baseline, HCV-RNA levels ranged from 12,000 to 2,410,000 IU/mL. Liver fibrosis was assessed using a noninvasive method, transient elastography (FibroScan) that that correlate the fibrosis stage as following: KPa ≤ 7.1 = F0−F1 (minimal fibrosis), 7.1 < KPa ≤ 9.5 = F2 (moderate fibrosis), 9.5 < KPa ≤ 14.5 = F3 (severe fibrosis), and KPa > 14.5 = F4 (cirrhosis) according to recent publications [14,15] (Table 1).

### 2.2. Serum Samples

Serum samples were stored at −80 °C until testing.

All specimens were tested for HCV-RNA levels by a commercially available method (ABBOTT GmbH & Co KG Max Plank-Ring-2, Wiesbaden, Germany); the detection limit was 12 IU/mL.

### 2.3. Amplification and NS3, NS5A, and NS5B Sequencing

HCV RNA was extracted as reported elsewhere using the Qiamp viral RNA minikit following the manufacturer’s instructions (Qiagen Viral RNA Mini Kit, Qiagen Strasse 1, Hilden, Germany) [16]. Then, synthesis and amplification of cDNA were carried out in a single step using the commercial Superscript III One-step RT-PCR system with Platinum Taq (Invitrogen by Life Technologies, 5791 Van Allen Way, Carlsbad, CA, USA). Primers for RT-PCR for each HCV genotype/subtype were designed from known sequences based on the NS3-protease (aa 1–181), NS5A domain (aa 1–213), and NS5B-polymerase (aa 1–591); if necessary, nested PCR was also performed with specific HCV genotype/subtype primers. Finally, NS3, NS5A, and NS5B amplified products were purified and sequenced by an automated DNA sequencing analyzer (ABI-3130) in sense and antisense orientation using the Big Dye Terminator Cycle Sequencing Kit v3.1 (Applied Biosystems, Foster City, CA, USA).

Wild-type amino acids were defined according to reference sequences from the Geno2Pheno HCV tool [17]. RAS detection and frequency analysis were performed by using the online informatics tool provided by Geno2pheno HCV. First, the nucleotide sequences from NS3, NS5A, and NS5B regions were analyzed by HCV BLAST in the Los Alamos HCV sequence database (http://hcv.lanl.gov/content/index) (accessed date: 2 September 2021). Then, the NS3, NS5A and NS5B sequences were aligned using the Clustal W algorithm integrated into the BioEdit software. Then, the sequences of HCV strains were aligned with a reference panel of sequences representative of each subtype (Genebank Accession numbers: Y11604, DQ418786; FJ462436; FJ462440), which is the same as proposed by Geno2pheno HCV tools [17]. The sensitivity for the detection of RASs using Sanger sequencing is approximately 10–20% [18].

Genotype and subtype was determined by phylogenetic reconstruction using PhyML 3.0 [19] under a GTR+G+I substitution model with NNI+SPR search and aLRT branch support to assess phylogenetic robustness. NS3, NS5A, and NS5B study sequences were aligned to the International Committee on Taxonomy of Viruses (ICTV) updated HCV genotype and subtype reference alignment version 8.5.19 (downloaded 7/21/21 from https://talk.ictvonline.org/ictv_wikis/flaviviridae/w/sg_flavi/57/hcv-reference-sequence-alignments) (accessed date: 2 September 2021)., which contains 223 reference sequences covering NS3, NS5A, and NS5B regions, using MAFFT version 7 [20]. Trees were visualized and annotated using the R package ‘ape’ [21]. For verification, subtype classifications were also performed using PhyloPlace (https://hcv.lanl.gov/content/sequence/phyloplace/) and HCV-Glue (http://hcv-glue.cvr.gla.ac.uk/) (accessed date: 2 September 2021).

The study was performed according to the Declaration of Helsinki, and ethical approval was obtained from Imam Abdulrahman Bin Faisal University (26 October 2020).

## 3. Results

Phylogenetic analysis showed the presence of different GT4 subtypes (Figure 1). Specifically, the GT 4a subtype was observed in five Saudi patients (patients 082, 263, 460, 589, and 710), in three patients from Egypt (585, 908, and 959) and one Sudanese patient (506). Three patients (234 and 532 from Saudi and 792 from Palestine) were infected with subtype GT 4d, and two Saudi patients (307 and 611) were infected with subtype GT 4o. Another two Saudi patients (119 and 517) were infected with apparent recombinant viruses consisting of GT 4a/GT 4o/GT 4a (in NS3/NS5A/NS5B). Interestingly, the recombinant virus in these patients appeared not to be directly epidemiologically linked as the closest relative reference sequences to the NS3 fragments were different, indicating that these recombinants were not closely epidemiologically linked. Furthermore, one patient from Saudi (141) was infected with an unclassifiable subtype of GT 4. This virus fell outside known GT 4 subtypes in all three genomic fragments analyzed. All subtype classifications, including both non-recombinant and recombinant assignments, were robustly inferred as indicated by high clade support (aLRT > 0.95), with the interesting exception of the unclassifiable sequence, which could not be robustly placed with any known subtype. The unclassifiable nature of this virus was also reported by the HCV-Glue online tool. Therefore, this sequence remains genotype 4 unclassified subtype.

In the NS3 region, the V170I mutation, associated with resistance to NS3 protease inhibitors, was detected in three patients, one infected with GT 4o, one with GT 4a, and one with apparent recombinant virus GT 4a/GT 4o/GT 4a. Several polymorphisms not associated with resistance to DAA treatment were found in all GT4 subtypes, as reported in Table 2. In particular, nine GT 4a and two GT 4a/GT 4o/GT 4a apparent recombinant patients showed A61S, S101A, A102S, I114V/L, I134T, R150A/V (reference sequence Y11604); three GT 4d patients had T95S or T95A (reference sequence DQ418786); two patients who harbored GT 4o showed, on the contrary, different patterns of mutations as shown in Table 2 with reference sequence FJ462440 (Table 2).

The sequence analysis of the unknown, unclassifiable, virus of GT 4 showed several polymorphisms compared to wild-type FJ462436. However, their significance is uncertain, as it is a single isolate.

In the NS5A region, two relevant mutations associated with resistance to NS5A inhibitors were found in two GT 4a-infected patients: the L28M in one case and L30R in the other one. In addition, the patient who harbored an unknown, unclassified virus displayed an association of both L28M and M31L mutations, which correlated with DAA resistance (Table 2).

In all GT 4 isolates, polymorphisms not associated with drug resistance were observed in the NS5A regions. In particular, K44R, V53M, K56T, and I99V/T were observed in all GT 4a isolates; D105N, D126E, A164P/T/Q, and L168M were present in GT 4d, although not in both viral isolates; M56I, E62N, K107E, I121V S127F, L158I, and C174S were found in GT 4o patients (Table 2).

No mutations associated with resistance were observed among NS5B sequences; however, also in this region, some polymorphisms were found according to the reference sequence. GT 4a showed amino acids changes at position 100, 130, and 213; GT 4d showed two changes at position 127 and 130, whereas in GT 4o, changes were observed at position 100 and 130 compared to sequence reference.

All patients were treated with a sofosbuvir (400 mg) plus daclatasvir (60 mg) regimen for 12 weeks. Three patients were lost to follow-up, whereas fourteen patients completed the treatment. A sustained virological response (SVR) was obtained in all but one who experienced a relapse. At baseline, a V170I mutation in the NS3 region was detected in the patient who relapsed. The patient was diagnosed using a noninvasive method by transient elastography for advanced fibrosis (F4); this patient was successfully retreated with glecaprevir (GLE) plus pibrentasvir (PIB).

## 4. Discussion

The study of genotype, subtype characteristics, and circulation is critical to defining HCV epidemiology and driving more appropriate therapy choices.

In the present study, GT 4 subtypes were assessed in 17 HCV GT 4-infected patients from Saudi Arabia. The most common subtype was GT 4a, and the other identified subtypes were GT 4o and GT 4d. Interestingly, two patients appeared to be infected with recombinant virus (4a/GT 4o/GT 4a), and one was infected with an unclassifiable virus, which may potentially represent a new, previously unseen, subtype. While the subtype classifications of all study sequences were statistically robust (aLRT > 0.95), meaning the mosaic structure was robustly inferred, we recognize that the exact recombination patterns cannot be revealed without full-genome sequencing followed by a detailed recombination analysis, which may represent a possible limit of the study. Likewise, the evolutionary history of the unclassifiable sequence may be revealed after full-genome sequencing and further analyses. This study further analyzed naturally occurring mutations in the NS3, NS5A, and NS5B regions associated with drug resistance.

The NS3 protease gene was successfully sequenced in all patients. Several reports demonstrated the presence of mutations associated with resistance to NS3 protease inhibitors; however, these studies were limited to GT 1 and GT 3 [7,8,9]. In the patients studied, we found the V170I mutation associated with resistance to NS3 protease inhibitors in three patients, of whom one was infected with GT 4a, one with GT 4o, and the other one was recombinant virus GT 4a/GT 4o/GT 4a. These results support a previous study [23], which reported a V170I in GT 4a isolates and at variance with another report in which the V170I was observed in GT 1a, GT 1b, and GT 3a [24]. This mutation has not yet been associated with resistance in the GT 4 isolates, but this could be due to the fact that this population is often under-represented in clinical trials [25]. Furthermore, the sequence analysis of the NS3 region of our HCV isolates showed the presence of other polymorphisms; however, these are not known as associated with DAA resistance.

Whether the polymorphisms which were observed in NS3 of Saudi patients could be associated with treatment failures is not known. In fact, it is possible that other unknown and less frequent mutations may confer a resistant phenotype. This makes the study of isolates before treatments crucial in order to optimize HCV infection therapy. All patients with V170I RAS at baseline achieved an SVR after treatment with SOF/DCV except one patient who relapsed, even though he was treated with a regimen not containing a protease inhibitor.

Paolucci et al. reported a mutation at position 168 in the NS3 region of GT 4a-infected patients with DAA failure [26]. Still, this mutation was not detected in any of our isolates.

Furthermore, the sequence analysis of the NS3 region of our HCV isolates showed the presence of several polymorphisms not associated with DAA resistance.

In the NS5A region, two clinically relevant RAS (L28M, L30R) were found in two patients harboring GT 4a. In contrast, in a patient with an unclassifiable virus, the presence of an association of L28M+M31L, was detected. In other non-GT 4 genotypes, these mutations have been associated with resistance to DAA [22]. Interestingly, the patients that harbored a mutation at position 170 in NS3 region did not show any mutations associated with resistance to NS5A inhibitors.

Paolucci et al. found neither L28M nor L30R associated with resistance to NS5A inhibitors in their HCV isolates; however, this study was only focused on GT 1a and GT 1b [27]. No mutations were observed in the NS5B region of our isolates in patients who harbored the V170I, either. However, several polymorphisms were found in this region compared to the reference sequence with no apparent clinical impact and were not associated with DAA resistance. In conclusion, in this small GT 4 cohort, most patients were infected with GT 4a. Although natural RASs were observed, they were not associated with DAA failure. Further studies on larger case series are needed to confirm these results. 

## Figures and Tables

**Figure 1 viruses-13-01832-f001:**
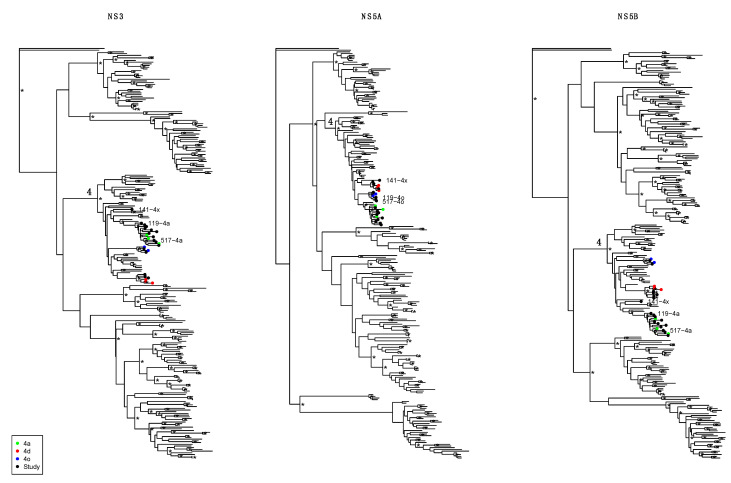
Phylogenetic classification of NS3, NS5A, and NS5B sequence fragments. Maximum likelihood phylogenetic trees show genotype and subtype inference of study sequences (“Study”, black symbols at tips) and ICTV reference sequences (unmarked tips and colored symbols of relevant subtypes; see legend). The genotype 4 clade is indicated by a “4” on the ancestral branch. Samples of special interest (recombinants 119 and 517, and unclassified 141) are indicated by name next to the corresponding black symbol. Phylogenetic robustness at aLRT > 0.95 is indicated by an asterix at the corresponding nodes. Trees were ladderized and rooted by a genotype 7a reference sequence (7a_EF108506).

**Table 1 viruses-13-01832-t001:** Clinical characteristics of 17 HCV Saudi infected patients.

Patients	Geographical Area	Sex	Age	Risk Factors	Fibrosis	Viral Load (IU/mL)
710	Saudi	F	61	Blood Transfusion	F1	2,410,000
959	Egypt	M	65	Unknown	F0	207,000
506	Sudan	M	32	Unknown	F2	181,000
585	Egypt	M	64	Unknown	Not available	302,000
908	Egypt	M	42	Intrafamiliar Transmission	F1	80,000
611	Saudi	F	66	Blood Transfusion	F4	743,000
517	Saudi	F	41	Blood Transfusion	F0	683,000
119	Saudi	M	36	Blood Transfusion	F0	130,000
307	Saudi	M	36	Blood Transfusion	F4	478,000
532	Saudi	M	35	Blood Transfusion	F3	1,940,000
263	Saudi	F	65	Surgery	F0	82,000
460	Saudi	M	32	Unknown	F0	153,000
589	Saudi	M	34	Blood Transfusion	F0	12,000
792	Palestine	M	73	Unknown	F4	239,000
141	Saudi	M	64	Unknown	F4	750,000
082	Saudi	F	59	Unknown	F4	495,000
234	Saudi	F	33	Blood Transfusion	F0	134,000

**Table 2 viruses-13-01832-t002:** NS3, NS5A and NS5B polymorphisms and naturally occurring resistance mutations to DAA in different clades of HCV GT 4 patients.

Patients	GT4 Subtype	NS3	NS5A	NS5B
710	GT 4a	A61S; S101A; A102S;	K44R; V53M; K56T; I99V;	N213T
		I114V; I134T; R150A	**L28M**	
959	GT 4a	A61S; S101A; A102S;	K44R; V53M; K56T; I99V	N213T
		I114V; I134T; R150A		
506	GT 4a	A61S; S101A; A102S;	K44R; V53M; K56T; I99V	N213T
		I114V; I134T; R150A		
585	GT 4a	A61S; S101A; A102S;	K44R; V53M; K56T; I99V	N213T
		I114V; I134T; R150A		
908	GT 4a	A61S; S101A; 102S; I114V;	K44R; V53M; K56T; I99V	N213T
		I134T; R150A; **V170I**		
611	GT 4o	L14F; V28A; T95A; S98T;	M56I; E62N; K107E;	R100K; A130T
		R149H	I121V; S127F; L158I;	
			C174S	
517	GT 4a/GT 4o/GT 4a	A61S; S101A; A102S;	M56I; E62N; I121V; S127F	N213T
		I114V; I134T; R150A;		
119	GT 4a/GT 4o/GT 4a	A61S; S101A; A102S;	M56I; E62N; I121V; S127F	N213T;
		I114V; I134T; R150A; **V170I**		
307	GT 4o	A61S; S101A; A102S;	M56I; E62N; I121V; S127F	R100K; A130T
		I114V; I134T; R150A; **V170I**		
532	GT 4d	T95S;	D105N; D126E, A164T;	R127L; T130N
			L168M	
263	GT 4a	A61S; S101A; A102S;	K44R; V53M; K56T; I99V	N213S
		I114V; I134T; R150V		
460	GT 4a	A61S; S101A; A102S;	K44R; V53M; K56T; I99V;	N213T
		I114V; I134T; R150A	**L30R**	
589	GT 4a	A61S; S101A; A102S;	K44R; V53M; K56T; I99T	N213T
		I114V; I134T; R150A		
792	GT 4d	T95A;	D105N; D126E, A164T;	R127L; T130N
			L168M	
141	unknown	A101S; T122N; S125A;	D126E; A164Q; L168M;	N62S; T131N;
		V151A; A166E	**L28M; M31L**	E202D; K212R
082	GT 4a	A61S; S101A; A102S;	K44R; V53M; K56T; I99V	N213T
		I114V; I134T; R150A		
234	GT 4d	T95A;	D105N; D126E, A164P;	R127L;
			L168M	T130I/V

Legenda: The polymorphisms associated with resistance to DAA according to the Geno2pheno HCV web system and Sorbo et al. 2018 [22].

## Data Availability

The sequences in this studies are available in GenBank database (2 September 2021).

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
