# Peer review of "Genetic Subtypes and Natural Resistance Mutations in HCV Genotype 4 Infected Saudi Arabian Patients"

_viruses, 2021, doi:10.3390/v13091832_

Round 1
Reviewer 1 Report
The authors have taken into consideration most of the reviewers comments.
However, I still think that recombination is a very rare event in HCV and full-length genome sequening has not been performed and attribution to rare subtypes 4 is difficult according to the database used for phylogeny.
Therefore the authours should not use the term recombination in the abstract and in the main text of the manuscript
Author Response
Reply to reviewers comments
However, I still think that recombination is a very rare event in HCV and full-length genome sequening has not been performed and attribution to rare subtypes 4 is difficult according to the database used for phylogeny.
Therefore the authours should not use the term recombination in the abstract and in the main text of the manuscript
REPLY: While recombination may be rare, we nevertheless observe these viruses in our study. The phylogenetic analysis robustly classified these viruses to different subtypes in the region studied, and in addition we found one virus that did not fall close to any known subtype. This was verified by HCV-Glue online service. We agree that we cannot reveal the precise recombination pattern without full genome sequencing, as stated in the discussion. That does, however, not change our statistically supported observation. To qualify our results, and acknowledge that we did not perform a whole-genome analysis, we have now used the term "apparent recombinants".
Reviewer 2 Report
Based on the review of the paper as requested, I think now it can be considered acceptable for publication and interesting for the readers of the Journal.
Author Response
We are grateful to the referee for the appreciation of the manuscript
Reviewer 3 Report
Dear Editor,
in this study, the authors characterized genetic subtypes of HCV4 in Saudi Arabian and analyzed prevalence and characteristics of resistance-associated substitutions (RASs) in DAA naïve patients. Additionally, they evaluated the impact of natural RASs on DAA treatment response. The overall presentation of brief report is good and well-focused. However, the manuscript needs some revisions:
- Introduction section, page 2, lines 63-64: The authors aimed to detect RAS. I suggest to specify RAS detection limit in the introduction section, maybe in the aim. Their approach, based on population sequencing, can detect variants around 15/20% of the viral population, as they report in Materials and Methods section.
- Materials and Methods section, page 3, lines 80-82: The authors report the liver fibrosis stage of enrolled patients. If available, I suggest to correlate the fibrosis stage as following: KPa ≤ 7.1 = F0−F1 (minimal fibrosis), 7.1 < KPa ≤ 9.5 = F2 (moderate fibrosis), 9.5 < KPa ≤ 14.5 = F3 (severe fibrosis), and KPa > 14.5 = F4 (cirrhosis) according to recent publications.
- Materials and Methods section, page 3, lines 106-107: The authors use BLAST to identify preliminarily HCV genotype. However, I suggest to use more specific tools, such as Oxford HCV Automated Subtyping Tool v.2.0 (http://dbpartners.stanford.edu/RegaSubtyping/html/subtypinghcvSUB.html) and COMET HCV typing tool (https://comet.lih.lu/index.php?cat=hcv).
- Materials and Methods section, page 3, lines 116-117: The authors perform phylogenetic analysis of NS3, NS5A and NS5B genomic regions. Before phylogenetic reconstruction methods, the authors should describe better the datasets construction for each region (i.e. total numbers of reference sequences included, the website used to download them with accessed date...) used to run analysis. Additionally, the references sequences and the 17 newly generated sequences should be indicated in the tree. Each branch tip should be linked to the corresponding sequence. Also, this is important to visualize the correlation (within cluster/clade) among risk factors, geographical area of origin and demographical data of enrolled patients.
- Results section, page 4, lines 141-142: “This sequence, therefore, remains genotype 4 unclassified subtype.”. Even if the HCV4 strain remain unclassified subtype, I guess the authors should report for each genomic region which cluster/clade include it by phylogenetic analysis.
Kind regards.
Author Response
Reply to Reviewer's comments
- Introduction section, page 2, lines 63-64: The authors aimed to detect RAS. I suggest to specify RAS detection limit in the introduction section, maybe in the aim. Their approach, based on population sequencing, can detect variants around 15/20% of the viral population, as they report in Materials and Methods section.
REPLY: RAS detection limits are now specified in the introduction aims sections as requested by the referee
- Materials and Methods section, page 3, lines 80-82: The authors report the liver fibrosis stage of enrolled patients. If available, I suggest to correlate the fibrosis stage as following: KPa ≤ 7.1 = F0−F1 (minimal fibrosis), 7.1 < KPa ≤ 9.5 = F2 (moderate fibrosis), 9.5 < KPa ≤ 14.5 = F3 (severe fibrosis), and KPa > 14.5 = F4 (cirrhosis) according to recent publications.
REPLY: Fibrosis stages are now correlated to KPa values according to Naqvi A et al. (2015), as suggested by the referee.
- Materials and Methods section, page 3, lines 106-107: The authors use BLAST to identify preliminarily HCV genotype. However, I suggest to use more specific tools, such as Oxford HCV Automated Subtyping Tool v.2.0 (http://dbpartners.stanford.edu/RegaSubtyping/html/subtypinghcvSUB.html) and COMET HCV typing tool (https://comet.lih.lu/index.php?cat=hcv).
REPLY: regarding this issue, we are grateful for the suggestion of the referee .
The phylogenetic analyses and BLAST, however, gave concordant results and we believe this makes data very trustworthy. We hope therefore that the referee will be satisfied of the methods we used.
- Materials and Methods section, page 3, lines 116-117: The authors perform phylogenetic analysis of NS3, NS5A and NS5B genomic regions. Before phylogenetic reconstruction methods, the authors should describe better the datasets construction for each region (i.e. total numbers of reference sequences included, the website used to download them with accessed date...) used to run analysis. Additionally, the references sequences and the 17 newly generated sequences should be indicated in the tree. Each branch tip should be linked to the corresponding sequence. Also, this is important to visualize the correlation (within cluster/clade) among risk factors, geographical area of origin and demographical data of enrolled patients.
REPLY: We have added a web link to the ICTV alignment, and made clear that the alignment contained 223 reference sequences in each region studied. The other comment appears to be in relation to the results: The trees in Figure 1 cannot be fully annotated with taxa names on every branch -- it would be too crowded. Therefore, as indicated in the legend, we have labelled all study sequences and relevant subtype reference sequences in the tree. In addition, the apparent recombinants and the unclassifiable variant has been highlighted by name. We feel this is the best visual representation of these data. However, we agree that the results presentation was not clear, and have therefore clarified the country of origin and identified each sample in the Results text. Refer to Table 1 for patient characteristics.
- Results section, page 4, lines 141-142: “This sequence, therefore, remains genotype 4 unclassified subtype.”. Even if the HCV4 strain remain unclassified subtype, I guess the authors should report for each genomic region which cluster/clade include it by phylogenetic analysis
REPLY: As detailed in the results the unclassifiable virus "fell outside known GT 4 subtypes in all 3 genomic fragments analysed" and the phylogenetic placement could not be robustly inferred by the aLRT test. Thus, it is not possible to assign any known subtype in any of the investigated genomic fragments. This virus remains unclassifiable at this time. As mentioned in the discussion, it may be possible to analyze this further, but that's outside the scope of this study.

Round 2
Reviewer 3 Report
Dear Editor,
I believe the manuscript has been sufficiently improved to be publish in Viruses.
Kind regards.
This manuscript is a resubmission of an earlier submission. The following is a list of the peer review reports and author responses from that submission.
Round 1
Reviewer 1 Report
In this paper, the authors aimed to determine the genetic subtypes of HCV-GT4 and identify the pre-existence of natural occurring resistance-associated polymorphisms in Saudi Arabia patients.
Major comments:
- Recombination: The authors stipulate that recombinant viruses were observed in 4/ 17 included patients (GT4a/GT-4o; GT4c/GT4d) based on divergent subtype attribution between the three sequenced regions (NS3, NS5A, NS5B) . This statement is over-interpreted. It is likely that subtypes related to these variants are not well matched in the database used for phylogeny. Despite high rate of genetic diversity in HCV, there has been little evidence that recombination plays a significant role in HCV evolution. Only one HCV circulating recombinant form has been discovered to date: a mosaic of subtypes 2k/1b. In addition, the breakpoints of intergenotypic recombinants have been detected in the nonstructural 2 (NS2) gene which makes recombination between NS3, NS5A and NS5B unlikely.
- Polymorphisms associated substitutions: Table 1 and table 2 are not clearly presented and confusing. The authors had better describe a unique table including all 17 patients, highlighting aminoacid substitutions known to affect resistance in NS3, NS5A and NS5B with for instance these positions in bold.
Minor comments:
- Introduction: P57-58 Other relevant recent papers should be cited that describe decreased virological response rates to DAA (i.e. Childs K et al. J hepatol 2019, Shah R et al . J hepatol 2021, Fourati S et al. Hepatology 2019, Nguyen D et al. J hepatol 2020)
- Regarding the patient who failed to SOF/DCV regimen, authors should clarify RASs at baseline (before SOF/DCV initiation) and after treatment failure.
- English should be edited/reviewed by someone who is fluent in English.
Author Response
RESPONSE TO REFEREE 1
MAJOR REVISIONS
We assumed that the viruses might be recombinant because both Geno2Pheno HCV and Phylogenetic analysis matched for different subtypes within the same strain of patients.
We accepted the suggestion of the referee: in fact now the results of naturally occurring mutations associated to resistance to DAA and polymorphisms in the three regions are in a unique table 2.
MINOR revisions
We added more recent papers as suggested
I am sorry but we got the plasma sample of HCV infected patient at baseline (before SOF/DCV initiation)

Reviewer 2 Report
In this study the authors present the phylogenetic analysis of NS3 NS5A NS5B sequences to characterize HCV subtype and natural RAS in 17 HCV infected Saudi Arabia patients with genotype 4. Moreover, the clinical outcome following daclatasvir/sofosfubir (12w) treatment in all patients was evaluated. The results shown the prevalence of subtype 4a (9/17) compared to other subtypes and 4 patients with recombinant subtypes 4a/4o (3/17) and 4c/4d (1/17). All patients but one achieved a sustained virologic response (SVR) to treatment.
Despite the small population studied, these preliminary data could contribute to improving our knowledge about the circulating HCV 4 subtypes, its recombinant strains and natural RAS, still poorly studied in Saudi Arabia. However, there are some issues that should be reviewed.
Material and Methods
- In the study population, fibrosis stage (F0-F4 ) informations could be added.
- It is not clear what is the method used for the RAS interpretation. Could you clarify in this section?
- Line 106 - now is fr
Results
- Figure 1 – In this figure should be indicate the genetic distance and bootstrap values to validate your results of subtyping and the probable intra-genotype (inter-subtype) recombinants identification. The image is not well defined and the identification numbers of the sequences are not clearly legible.
- Could you please describe the recombinant viruses identified in more detail? For example, how are the combinations in NS3 NS5A and NS5B regions with different subtypes in the same strain? Althought a small population, I find interesting the frequence of the recombinant subtypes (4 out of 17 patients). Recombination in HCV is a rare event and the number of well-documented cases is still very low.
- Table 1 – In this table should be indicate the polymorphisms resistance-associated according to the guidelines used (for example: geno2pheno HCV web system, Sorbo et al 2018, EASL guideline and/or other reference) and consequently modify the table or its legend. As also specified in the discussion (line 171), the V170I polymorphism is not considered resistance-associated in HCV 4 genotype. Unfortunately I did not find the reference n.14 (line 248) in pubmed which perhaps describes the potential role of the V170I natural polimorphysm in HCV 4 genotype. The reference n.15 (line 251) describes the presence of the V170I as natural polimorphysm in genotype 1a 1b and 3a but not in Gt4. Do you know other references in literature about it?
- “natusubtype” should be correct as “subtype”
- Line 150 - It is certainly irrelevant to associate the therapeutic failure DCV/SOF (NS5A and NS5B inhibitors, respectively) with the presence of V170I natural polymorphism in NS3. Analysis of the NS5A and NS5B sequences at failure could detect any RAS acquired, do you have these data? Could you describe important clinical data of the patient who failed the therapy? (liver fibrosis, interferon experience, treatment adherence …).
Based on these observations the authors should review some considerations in the results and discussion of this paper. This study could make a better scientific contribution in this area by adding some information and taking care of some important details.
Author Response

(The authors gave the same response as above.)

Round 2
Reviewer 1 Report
One important issue is recombination. The authors stipulate that recombination were observed in 4/ 17 included patients (GT4a/GT-4o; GT4c/GT4d) based on divergent subtype attribution between the three sequenced regions (NS3, NS5A, NS5B). I believe this statement is over-interpreted. It is likely that subtypes related to these variants are not well matched in the database used for phylogeny (the database can miss some subtypes sequences and therefore subtypes are not accurate). Despite high rate of genetic diversity in HCV, there has been little evidence that recombination plays a significant role in HCV evolution. Only one HCV circulating recombinant form has been discovered to date: a mosaic of subtypes 2k/1b. In addition, the breakpoints of intergenotypic recombinants have been detected in the nonstructural 2 (NS2) gene which makes recombination between NS3, NS5A and NS5B unlikely. Full-length genome sequencing is required with specific bioinformatic recombination softwares to stipulate that recombination occurs.